# Agent-Based Approach for (Peri-)Urban Inter-Modality Policies: Application to Real Data from the Lille Metropolis

**DOI:** 10.3390/s23052540

**Published:** 2023-02-24

**Authors:** Azise Oumar Diallo, Guillaume Lozenguez, Arnaud Doniec, René Mandiau

**Affiliations:** 1Laboratoire Aménagement Économie Transports (LAET), ENTPE-CNRS-University of Lyon, 3 Rue Maurice Audin, 69120 Vaulx-en-Velin, France; 2CERI Systèmes Numériques, IMT Nord Europe, 59650 Villeneuve d’Ascq, France; 3Univ. Polytechnique Hauts-de-France, CNRS, UMR 8201-LAMIH, 59313 Valenciennes, France

**Keywords:** agent-based modeling, utility-based agent, multinomial logit model, inter-modality, *MATSim*

## Abstract

Transportation authorities have adopted more and more incentive measures (fare-free public transport, construction of park-and-ride facilities, etc.) to reduce the use of private cars by combining them with public transit. However, such measures remain difficult to assess with traditional transport models. This article proposes a different approach: an agent-oriented model. To reproduce realistic applications in an urban context (a metropolis), we investigate the preferences and choices of different agents based on utilities and focus on a modal choice performed through a multinomial logit model. Moreover, we propose some methodological elements to identify the individuals’ profiles using public data (census and travel surveys). We also show that this model, applied in a real case study (Lille, France), is able to reproduce travel behaviors when combining private cars and public transport. Moreover, we focus on the role played by park-and-ride facilities in this context. Thus, the simulation framework makes it possible to better understand individuals’ intermodal travel behavior and assess its development policies.

## 1. Introduction

Faced with the issues of global warming and the high cost of living, political actors are increasingly looking at collective and less polluting transport systems (e.g., bike sharing, carpooling, and inter-modality transport) to provide solutions that are both inexpensive (for the customer) and ecological. These policies, for the most part, aim to reduce the use of private cars in favor of soft modes of transport (e.g., bicycles and walking) and public transport (*PT*). They generally consist of adjusting public transport pricing [1], accelerating the energy transition (e.g., development of electric vehicles), and developing transport and urban infrastructure (e.g., park-and-ride facilities and multi-modal interchanges).

In France, for example, such policies are included in the mobility orientation strategy, which aims to strengthen the transport supply by (1) rebalancing the modal shares in favor of public or active modes (walking and bicycles), (2) strengthening the shared use of individual transportation modes (carpooling and car sharing), and (3) encouraging multi-modality and inter-modality. Inter-modality can be defined as the combination of several transportation modes during a single trip [2]. To a certain point, it responds to the problems of congestion, parking, and gas emissions [3]. Furthermore, its practice nowadays is facilitated by the development of new mobility offers and the availability of new information technologies that ease the planning of trips involving different transportation means. The most common form of inter-modality consists of combining private vehicles with public transport (*PV*+*PT*) [4]. It often relies on using park-and-ride facilities which are usually located at public transport stations. The capacity, location, and cost of using these infrastructures are among the many factors influencing inter-modal *PV*+*PT* policy [5].

Therefore, it is essential to think carefully about decisions to promote this practice. To evaluate their policies, mobility and transport stakeholders often use software for traffic simulations [6]. Different levels of modeling depend on the degree of granularity that is possible: macroscopic [7], mesoscopic [8], or microscopic [9,10]. A simplified vision of these viewpoints could be translated into two main families for transportation simulations: flow-based or agent-based models. In a flow-based model, each moving entity is seen as a physical particle in a flow or a force field. This modeling is particularly relevant for studying the traffic flow on a highway or the size of a public transport line. Physical constraints (e.g., minimum distance or dynamics based on velocities and accelerations) take precedence over decision-making aspects. The other point of view considers that the decisions result from the cognitive processes of the different active entities (called agents). The agent perceives and acts on the environment according to its knowledge and preferences. Thus, it can be based on mobility simulation tools and, in particular, agent-based approaches [11], which are well-suited to the complex nature of transport systems [9]. This approach, called the *agent-based model* (*ABM*), describes a system as a set of agents interacting with each other. The *ABM* is particularly distinguished by the study of the decision-making processes underlying the behaviors, because there is a natural relation between the individuals (from a transportation system) and the agents (from our model). To our knowledge, the inter-modal behaviors within these tools are not very well defined or are even missing [12,13].

Our approach describes a multi-agent simulation to reproduce inter-modal trips from publicly available mobility data. The type of inter-modality studied concerns the combination of private cars and public transport (*PV*+*PT*), which requires park-and-ride facilities (*PR*). The modal choice is defined by a multinomial logit (*SI-MNL*) model coupled with the simulation tool *MATSim*/*eqasim* [14,15]. Note that combining the agent-based simulation with a discrete choice model is difficult for reproducing realistic applications [16,17].

We show, step by step, how to build the missing information from the data, which is necessary for estimating the discrete choice model and creating the transport demand. Then, to allow the practice of the *PV*+*PT* inter-modality in *MATSim*, we develop a new alternative with a specific routing approach based on park-and-ride facilities.

The developed simulation framework is applied to the European Metropolis of Lille (*MEL*, a french acronym for *Métropole Européenne de Lille*) in France. After calibrating and validating the simulation with household travel survey data, we evaluate the use of the *PR* facilities and study the agents’ behavior using this alternative. The simulation shows that some *PR*s located at the entrance to Lille are overloaded during peak hours. The simulation framework proposed can thus be used as a tool to explore actions aimed at promoting the combination of private cars and public transport, such as the construction of new *PR*s and free access to public transport subscribers.

The remainder of this article is organized as follows. Section 2 presents the background concerning the available data sources and the approaches for modeling and reproducing the multi-modal transport, particularly through the *ABM* based on utilities (the modal preferences and choices). Section 3 presents our approach: how anonymized data from multiple sources can be used to create consistent artificial populations and the integration of inter-modal behaviors (*PV*+*PT*). Then, Section 4 evaluates the results based on the data from the European Metropolis of Lille (*MEL*) and validates our approach. Finally, Section 5 presents a discussion, and Section 6 gives our conclusions and perspectives.

## 2. Background

First, we describe the data and how they can be used today to simulate travel demands for a region (Section 2.1). Then, we briefly introduce transport system simulation approaches, which are mainly those based on agent-based models and dealing with inter-modal practices (Section 2.2).

### 2.1. Available Data Sources

Agent-based approaches require huge amounts of data to allow the reproduction of the studied population. In the following, we present the data sources available in France, which are useful for *ABM* mobility demand. They can be grouped into two categories based on the acquisition method. Thus, we distinguish the active approach, where data from surveys are gathered through questionnaires addressed directly to individuals (Section 2.1.1). The second method, called the passive approach, consists of collecting data from sensors or other data acquisition devices (Section 2.1.2).

#### 2.1.1. Data from Surveys

In France, the main data sources on mobility come from the census and household travel (*EMD*, a french acronym for *Enquête Ménage Déplacement*.) surveys.

The population census provides information on individuals’ socioeconomic characteristics (e.g., age, socio-professional categories, and households). In France, these data characterize 35% of the population. Since it does not concern people’s travel habits and mobility behaviors, the census does not provide enriched travel information but only the main activities (work and education).

Based on a census (around 0.8–2% of the population), the *EMD* is the unique survey providing mobility practices. The survey considers the detailed reasons (primary and secondary activities), the number of trips, the intermediate trips, and the transportation mode actually used. The data also contain some main users’ socio-economic characteristics (e.g., age and gender). These are often used to establish public mobility policies.

Traditionally, travel surveys and the population census are the data used for agent-based transport demands [18,19]. New data sources from new information and communication technologies (e.g., sensors and smartphones) also provide information on mobility practices.

#### 2.1.2. Data from Sensors

Obtaining mobility data from the sensors can be defined as a passive method and classified into two groups: massive data from GPS and smartphones and data from cameras and induction loops.

Massive mobility data are mainly derived from *floating mobile data* (*FMD*) and *floating car data* (*FCD*). *FMD* are obtained either by triangulation or by the inter-cellular transfers and the relay antennas of the operators during travel. *FCD* are generally obtained from vehicle location data (*GPS*). They can also be obtained via the locations of phones when they are switched on in a vehicle. These data provide better knowledge of the O/D flows than data from standard surveys, as the information is more detailed and more recent for users’ travel routes due to the frequently updated location system. These data are handy for calibrating the model [20] and estimating the travel demands for public transport [21]. *FCD* and *FMD* are obtained through their use.

The authorities organizing mobility are deploying more and more devices (sensors) to control user access to public transport and parking facilities by validating tickets and subscription cards or capturing Wi-Fi signals. The volume of captured data provides crucial information for sizing the transport offer according this demand. Thanks to ticketing data, it is possible to build travel O/Ds [22], perform short-term forecasts, or measure the impact of external events [23]. Wi-Fi data are used, for example, to monitor the flow of public transport users [24].

The main limitation for their processing and use is the lack of trip data (e.g., trip purpose, modes of transportation used, and individual characteristics). They can also come from the operator’s facilities (telephone or satellite and card readers), but they may not represent the global population (for example, no data on non-subscribers).

Among all these data sources, only *EMD* provide information on inter-modality behaviors through the description of journeys and the different transport modes during the trip. Ticketing data could also provide information on the combination of public transport with other modes, particularly the private car (through relay car parks) or shared modes such as bicycles and electric scooters. However, to extract such information would require an in-depth analysis of these data.

### 2.2. Approaches for *ABM* Simulation

Recent developments in simulation tools allow for performing flow- or agent-based studies of transportation systems. We introduce the activity-based model (Section 2.2.1). Then, we recall transportation demand and supply generation (Section 2.2.2) based on the *ABM*. Finally, we focus on the modal choice supporting the utilities for the agents (Section 2.2.3).

#### 2.2.1. Activity- and Agent-Based Model

The individual practices consider the motivations and the different activities for all their trips (from and toward his or her home). In contrast to a traditional (macroscopic) model [25,26,27], the *activity-based* approach deals with the selected transportation modes during the trip while ensuring the consistency of the combination of modes (e.g., to take the vehicle if the mode for public transport is unavailable) [28,29].

In addition, the activity-based model allows, through synthetic population generation techniques, evaluating different mobility policies in a more straightforward fashion than with a macroscopic model. Combining activity-based and agent-based models is a way of reproducing the interactions between the individuals during their performed daily activities [30]. Thus, the individuals’ decision making (modal choice) is modeled more explicitly. In the following, we recall the *ABM* by considering the modal choices.

Earlier, a comparative study of the seven most popular *ABM* simulators was already proposed in [13]. The work showed that *MATSim* [14] and *SUMO* [31] are particularly suitable for studying mobility policies. Moreover, *MATSim* reproduces the inter-modal behaviors better through an activity-based approach and the notion of utility for the agent’s decision process (see Section 2.2.3). *MATSim* has been successfully used in many mobility studies. To the best of our knowledge, there is only one work on inter-modality using *MATSim* [32]. This inter-modality only concerns the combination of public transport (*PT*) with shared micro-mobility (bike) services. However, our study focuses on the combination of private cars with *PT* through the use of park-and-ride facilities.

#### 2.2.2. Multi-Agent Transportation Demand and Supply Generation

Transportation demand is typically derived from a synthetic population. It is a collection of agents with attributes such as home location, age, gender, socio-professional categories, and ownership of transport data (e.g., license, car, and public transport ticket or pass). The transportation demand includes the activities and trips performed by the agents. It includes elements such as the types of activities and their locations, the beginning and ending times for these activities, and the trip mode. Thus, it is possible to create more refined analyses by considering the agents’ individual characteristics.

To the best of our knowledge, there are no normalized data for the population or transport demand. Demand generation must be performed by the modeler or researcher based on the available data. Some *open source* synthesizers for different case studies are available [19,33]. Nevertheless, adaptation efforts are necessary to correspond to the study area.

#### 2.2.3. Modal Choice for the Decision Making

Several factors related to the individuals (e.g., socioeconomic category, gender, age, and trip purpose) and the transportation mode (e.g., trip duration, cost, and travel comfort) have been identified as impacting the modal choice [4]. Mathematical models of discrete choice [34] are then used to describe the modal choice for each individual. They are based on the concept of the utility that an individual *i* has for an alternative *m* (transportation mode in our case study) [35]. This utility is a function of the individual’s characteristics (e.g., professional category and age) and the attributes of the alternative (e.g., time and cost for a trip), to which we add a random error.

The notion of utility allows for a model that is more realistic for the individuals’ travel behaviors (i.e., the choice of their transportation mode). Individuals will choose a transportation mode or a combination of modes, maximizing their utility. The modal choice can thus be easily taken into account by the agents. This naturally leads to considering the approach described by the *utility-based agent* [36]. Through an iterative approach, agents can thus learn from their activities (actions) to change their choices.

## 3. Methodology and Assumptions

Figure 1 summarizes our approach for performing an agent-based inter-modality simulation. We can assume that the mobility demand and the transport supply are known for the studied area. These data (introduced in the previous section) are necessary for our approach. The different modeling steps are detailed below.

First, we explain the choice of the *MATSim* platform (Section 3.1) and how to define trip (particularly inter-modal) behaviors for each agent (Section 3.2).

### 3.1. Our Choice for Inter-Modal Simulations Based on MATSim

This work is based on the Multi-Agent Transport Simulation (*MATSim*) software [14] coupled with an add-on module for the modal choice (called *eqasim*) [15]. Figure 2 describes the general simulation loop for *MATSim*. Each individual (agent) has a daily *plan* of trips and activities. Agents’ initial daily plans are then represented by the **initial demand** with the transportation supply (e.g., the road network and the *PT* network). The plans of all agents are defined through the mobility micro-simulation module *QSim*, generating the associated flows and also different phenomena such as traffic congestions and emissions. The plans are then evaluated by a score based on the performed activities and the utility provided by the transportation mode. At each iteration, a subset of the agents can modify their plans to improve their score by changing either the transportation mode, the route, or the departure time (**Re-planning**). Each agent, in turn, will have the possibility to optimize his or her daily mobility plans. A co-evolutionary algorithm is used to reach an equilibrium state where no improvement is possible.

In *eqasim*, the initial decision-making module is replaced by a discrete choice module (**DMC**). Therefore, *eqasim* allows better behaviors by focusing on the modal choices using a discrete choice model and does not enable other replanning strategies. The simulation produces new trips captured by a **Variables** component (e.g., travel times and distances) for the modal choice module. These new values, resulting from interactions between agents, may vary from one iteration to another (e.g., presence or absence of congestion). **Estimators** compute the utility of different modes based on their variables and parameters.

The current version of *MATSim*/*eqasim* requires as input the origin/destination (O/D) coordinates of the activities for the agents’ plans. However, to avoid indirect identification of persons in the surveys, the areas (municipalities) of residence, their activities, and the O/D locations of their trips are deleted. Moreover, the data on inter-modal practices are not directly usable without preprocessing. We show how to rebuild these data without compromising the privacy and how to extract the information on inter-modality.

### 3.2. Rebuilding the Anonymized Data

We detail the rebuilding for the O/D coordinates (Section 3.2.1) and inter-modal trips (Section 3.2.2) from the available data sources.

#### 3.2.1. O/D Coordinates Design

Survey data are commonly based on predefined zones, (drawing sectors) to maintain the anonymity of individuals. No information on the municipalities of residence or activities (work, school, or shopping) is thus provided. However, these sampling areas must be consistent with the administrative divisions while ensuring scaling of the results for population of this area. The information on the municipalities is necessary for the generation of the synthetic population in order to have observations from the studied area.

To define the municipalities of residence and activities, we superpose the shapes of the drawing sectors (usually provided with the survey data) on those of administrative divisions (usually accessible). The drawing sectors, which are generally smaller, will thus appear for the corresponding municipalities. This task can be realized with software for geographical data processing, such as *QGIS* and *ArcGIS*.

In addition, the O/D coordinates of trips are necessary to estimate the discrete choice model and also for the creation of the activity plans for each agent. Thus, we aim to redefine consistent coordinates from the O/D drawing sectors and Euclidean distances provided in the survey. We propose a method which can be defined by two steps: (1) generating random points (N∈N) in the O/D drawing sectors and (2) searching for a point from the origin area corresponding to a point in the destination area so that the (Euclidean) distance between these two points is the closest possible to that provided in the survey. Note that we keep the point coordinates corresponding to the home and ensure the consistency of trips between the home and other activities.

#### 3.2.2. Inter-Modal Trips Design

*EMD* data are grouped into four files specifically addressing the household, person, trip, and travel information. The *household* file provides information on all household characteristics (e.g., motorization, location, and type of housing). The *person* file provides information on characteristics such as age, occupation, and driver’s license ownership. The *trip and leg* files provide information on the sequence of the transportation mode used, trip purpose, departure time, and distance. It is possible to recover the inter-modal trip information based on the *trip* file. Note here that we consider, as an inter-modal trip, the combination of several mechanized transportation modes (vehicle, public transport, and bicycle), even if some studies also consider the combination of walking (from a certain distance or walking time) and public transport as an inter-modal practice [38]. Thus, the combination of several public transport (*PT*) modes (e.g., bus and metro during the same trip) is an inter-modal trip [4,38].

To rebuild the inter-modal trips, we identify all trips for each trip by merging these two files (*trip and leg*). In addition, the modelings consider the following transportation modes:All public transport modes (bus, subway, tramway, and train) are represented by a single *PT* mode. This aggregation of PT modes is needed to enforce this alternative compared with the other transportation modes (car, walking, and bike). In addition, we currently focus on an inter-modality combining public transport with other individual transportation modes (car and bike). The public transport modes, therefore, are merged to facilitate identification of the combinations with the car and the bicycle and to have enough observations of this practice to carry out the simulations.The car mode, as the driver or a passenger, is considered as a private vehicle (*PV*). For example, a driver taking a private vehicle and then public transport will be considered as *PV*_driver+*PT*=*PT*+*PV* (a similar way for *PV*_passenger+*PT*=*PT*+*PV*).For the inter-modal combinations with public transit, we consider all *PT* transfers as one during the trip. For example, a person taking a private vehicle and then two public transport modes will be considered as a private vehicle and a public transport (*PV*+*PT*+*PT*=*PT*+*PV*).It is also assumed that we do not consider combinations that are too unrepresentative, such as *PV*+*Bike* or *PV*+*Bike*+*PT*. We also do not consider the combination of *PT* (*PT*+*PT*) modes in this study.

The following section describes the inter-modal behaviors of each agent, particularly the modality based on private vehicles and public transport.

### 3.3. Inter-Modality Approach Based on a Private Vehicle and Public Transport

The modal choice options propose different alternatives via *eqasim*: private car, public transport, bicycles, and walking. Therefore, the alternative, *PV*+*PT*, should be explicitly defined to allow inter-modal choices. We propose new transportation modes allowing this practice (Section 3.3.1) and describe its conditions of use and its routing module (Section 3.3.2). We also define the decision making based on utilities for the agents (Section 3.3.3).

#### 3.3.1. Principle of the Inter-Modal Alternative (*PV*+*PT*)

The combination of private car and public transport (*PV*+*PT*) should be performed through infrastructures, particularly park-and-ride (*PR*) facilities, as illustrated in Figure 3.

We propose two new complementary transportation modes: (1) *car_pt* for the outward journey and (2) *pt_car* for the return journey. The *car_pt* alternative (shown in red) is defined by a private vehicle (as the transportation mode) from the home to the *PR* (first trip) facility. The second trip is performed by the *PT* mode (e.g., bus or subway) from the *PR* to the destination, or more precisely to the nearest station, followed by an egress of walking. The same process is performed for the return trip, using the alternative called *pt_car* (blue). In this case, the first trip is made by public transport from the previous destination to the same parking facility used for the outward journey. In the second trip, the individual returns home using the same vehicle left at this parking facility.

#### 3.3.2. *PV*+*PT* Usage Constraints and Routing Approach

In order to ensure a consistent generation of possible cases for the *PV*+*PT* alternative, we define four constraints, which are presented as follows:Only agents owning a car and a public transport ticket can use the *PV*+*PT* alternative.The same *PR* facility must be used in the outward and return journeys.The home must be the origin (destination) of the trip performed by *PT*. In this case, the car must be picked up and dropped off at the agent’s home.The chain of alternatives, including the modes *car_pt* and *pt_car*, are only considered during the modal choice process.

*MATSim* assumes that the transportation mode’s routing process must be defined explicitly. Otherwise, users using these modes will be directly moved to their final positions. Moreover, the current version of MATSim/eqasim cannot simulate the combination of a private car and PT via PR. Figure 3 shows that this routing for the *car_pt* and *pt_car* modes depends only on the used park-and-ride (*PR*) facilities. Therefore, the *PR* choice must be achieved before the routing. The choice of park-and-ride facilities is then defined as the decision to select one among a set of *PR*s located close to the agent’s house. The process of searching for a parking space and the capacity of the *PR* are not taken into account for the moment to propose a straightforward model. In addition, the choice of using the *PR* closest to the individual’s home is also relevant under current mobility policies and aims to favor public transport rather than private cars. This modeling choice is also justified by the unavailability of *PR* usage data to identify individual preferences when considering elements such as the search times, the parking costs, and the walking times [39]. The routing module calculates the route by car from a home to the selected *PR*. Then, it performs *PT* routing from the *PR* to the closest public transit stop (station) for a given destination. The rest of the trip is performed by walking to reach the destination (the workplace, in this example). We propose a travel routing algorithm for *PV*+*PT* as follows:1.Identify the *PR* closest to the agent’s home.2.Perform routing by car between the home and *PR* using the routing module proposed by *MATSim*.3.Carry out the routing between the *PR* and the destination using the public transport routing module. This can be also achieved by using the routing module.4.For the return journey, the analysis is similar but with a starting time for the trip by public transport.

Thus, during the simulation, the routing module of *PV*+*PT* receives the following information: the characteristics of the agent (e.g., age, car ownership, and home location) performing the trip, the O/D of the trip, and the location of the park-and-ride facility (provided by the *PR* search module), which are necessary to calculate the route.

#### 3.3.3. Multi-Nomial Logit Model for the Inter-Modal Alternative

The two transportation modes (*car_pt* and *pt_car*) representing the inter-modal choice (*PV*+*PT*) have the same characteristics for the trip, except for the routing, as mentioned above. A multinomial logit model (*MNL*) called the simple inter-modal *MNL* (*SI-MNL*) is used to design the modal choice.

The *MNL* is justified by the simplicity of their set-up [34,40]. These models are based on an essential property, namely the *Independence of Irrelevant Alternatives* (*IIA*, a property of independence with respect to the other alternatives). Therefore, we suppose that the modal choices are independent, and the error terms are without correlations. The hierarchical structure of the model is presented in Figure 4. All the modes are at the same level of consideration. The utility function (Equation (Equation 1)) representing the inter-modal alternative is based on the *PV* and *PT* functions (Equations (Equation 2) and (Equation 3), respectively): (1)$$\begin{array}{c} U_{i,car\_pt}=\beta_{ASC,car\_pt}+U_{i,PV}-\beta_{inVehicleTime,PV}\times \theta_{parkingSearchPenalty}\\ -\beta_{accessEgressWalkTime}\times \theta_{accessEgressWalkTime}+U_{i,PT}-\beta_{ASC,PV}-\beta_{ASC,PT} \end{array}$$
(2)$$\begin{array}{c} U_{i,PV}=\beta_{ASC,PV}+\beta_{inVehicleTime,PV}\times \left(x_{inVehicleTime,PV}+\theta_{parkingSearchPenalty}\right)\\ +\beta_{accessEgressWalkTime}\times \theta_{accessEgressWalkTime}+\beta_{cost}\times x_{cost,PV} \end{array}$$
(3)$$\begin{array}{cc} U_{i,PT} & =\beta_{ASC,PT}+\beta_{inVehicleTime,PT}\times x_{inVehicleTime,PT}\\ \; & +\beta_{accessEgressTime,PT}\times x_{accessEgressTime,PT}+\beta_{numberTransfers}\times x_{numberTransfers}\\ \; & +\beta_{transferTime,pt}\times x_{transferTime,PT}+\beta_{cost}\times x_{cost,PT} \end{array}$$
where the following definitions apply:*i* corresponds to the individual, agent, or group of individuals;β represents the parameters (coefficients) of the model to be estimated. They determine an alternative’s utility (value) for a given individual during the simulation.*x* represents the variables corresponding to the known individual’s characteristics (e.g., age and gender) and the alternative’s attributes (e.g., travel time and cost).The *alternative specific constants* (*ASC*) define the variation of choice not explained by the known attributes exclusively.θ represents the calibration parameters to be adjusted manually during the simulation. These parameters make it possible, for example, to consider the constraints of use (search or cost for parking) of the private car, which are not provided in the travel survey data.

In the following section, we present the results of different experiments performed to validate our approach.

## 4. Our Case Study

This section presents an application of the developed framework. We describe the studied area and the simulation set-up details (Section 4.1), as well as the obtained results (Section 4.2).

### 4.1. Description of the European Metropolis of Lille (MEL)

The *MEL* is composed of 95 municipalities with 1.1 million inhabitants over an area of 672 km^2^. As in most European metropolises, we observe that the population is gathered in Lille (the main city), while the rest of the metropolis is weakly populated. Note that the number of vehicles per household is higher in these peripheral areas, whereas around 70% of households in the center own a private vehicle.

The public transport network includes two metro lines (43.6 km), two tram lines (22 km), and around 90 bus lines. There are currently 12 *PR* facilities, corresponding to 5005 parking spaces. For our study case, we use the *EMD* (https://opendata.lillemetropole.fr/explore/dataset/enquete-deplacement-2016/information/?location=10,50.65641,3.03338&basemap=jawg.streets, accessed on 14 December 2022), carried out in 2016, which informed us of the recent mobility practices of the people. The vehicle (44% as a driver and 14% as a car passenger) is the primary transportation mode with a score of 58%, followed by walking (29%), public transit (7%), bikes (2%), and inter-modal combinations (4%), totaling 16% of *PT*+*PT* trips).

### 4.2. Results

We present here the parameters of the discrete choice model (Section 4.2.1) and the validation of the results (Section 4.2.2 and Section 4.2.3).

#### 4.2.1. Discrete Choice Model (*SI-MNL*) Parameter Estimation

Table 1 presents the estimated parameters of the model using one-third of the data from *EMD* 2016 of *MEL*. The model was estimated with *PandasBiogeme* (the last version of the open source package *Biogeme* for discrete choice model estimation, and commonly used in transportation modes) [41].

The estimated model (*SI-MNL*) was statistically significant, with an initial log likelihood (LL) equal to −8668.555, while the estimated LL was −5086.908. The initial LL was estimated only with constants. The improvement in the LL values for our *SI-MNL* model shows that the explanatory variables introduced were significant. The significance of the estimated model was also confirmed by the rho square equal to 0.412, which is a high value. Moreover, the estimated parameters were statistically significant, except those related to the duration of time spent in the car ($\beta_{inVehicleTime,PV}$) and *PT* ($\beta_{inVehicleTime,PT}$) and the time to reach the *PT* ($\beta_{accessEgressTime,PT}$, $\beta_{numberTransfers}$, and $\beta_{transferTime,PT}$), with a p-value greater than 0.5.

The quality of our model was also evaluated through the signs of the parameters. The sign of $\beta_{time}$, except for the *PT*, was negative, reflecting the *disutility* of these transportation modes. The marginal utility of the spent time for the *PT* was positive, because commuters can, for example, achieve something else during their journeys. For example, they perceive *PT* modes as safer than cars and bicycles. However, the time spent during the transfer negatively impacted this mode. A similar analysis showed that the $\beta_{cost}$ was negative, as the more an alternative has a high cost of use, the less attractive it will be.

Once the values of the parameters were estimated, the model could be used in the simulation to predict the mode choice. Thus, (manual) calibration was then carried out to reproduce the reference situation according to the household travel survey.

#### 4.2.2. Calibration and Validation of the Simulation

The initial travel demand (including the agents’ daily plans) and supply were generated by adapting the synthesizer pipeline, initially developed for Ile de France [19]. All the data used (in particular the *EMD*, public transit supply, and the park-and-ride facilities) to carry out the simulation are publicly accessible on the open data site of the *MEL*. Due to computational time-saving reasons, 20% of the actual population, corresponding to 225,240 agents and 746,318 trips, was simulated. The travel supply (road and transit capacity) was reduced accordingly for the sample. The simulation was executed for 300 iterations, with 20% of agents able to change their transportation modes. The computation time was 10 h on an Intel^®^ core™ i7 (8 CPUs) processor with 16 GB of RAM.

The coupling between a discrete choice model and the multi-agent simulation was more efficient for the convergence of different results, particularly for the estimated parameters. However, the simulation may not have reproduced the reference situation exactly. It was therefore necessary to calibrate the model by adjusting certain specific parameters. For example, we adjusted the parameter of the *PV*+*PT* alternative ($\beta_{ASC,acr\_pt}$) because it was not really estimated in the discrete mode choice model.

Figure 5 presents four obtained cases representing the modal shares of different alternatives for different *ASC* values for *PV*+*PT*: $\beta_{ASC,car\_pt}=0$ (Figure 5a), $\beta_{ASC,car\_pt}=10$ (Figure 5b), $\beta_{ASC,car\_pt}=2$ (Figure 5c), and $\beta_{ASC,car\_pt}=1.25$ (Figure 5d). These results illustrate the obtained results from a set of numerous simulations carried out during the calibration of our model. They give an idea of this time-consuming manual calibration process from extreme values (Figure 5a,b) until the obtained values producing better outputs (Figure 5c,d). The simulation was subsequently validated by comparing the travel times and distances.

The modal shares for each mode (alternative) are presented on the ordinate axis, while the Euclidean trip distances are on the abscissa. This representation allows a better appreciation of the modal distributions. For example, we can see that the walking share was very large for short distances and close to zero for long distances, corresponding to our intuition. Case 4 (selected for the following results), with $\beta_{ASC,car\_pt}=1.25$ (Figure 5d), produced a simulation output similar to the reference situation. There were some differences for long trips (more than 12 km) realized by *PV*, *PT*, and *PV*+*PT*. This difference was mainly due to the lack of baseline observations of these types of trips, which were, in our context, mostly occasional.

Figure 6 gives the average distribution for the travel distances (Figure 6a) and travel durations (Figure 6b) (with or without the inter-modal *PV*+*PT* alternative) from the household travel survey.

The distances and durations from the simulation corresponded thoroughly with the observed data from the *EMD*. However, there was a slight variation between the simulated and observed travel times. This difference can be explained mainly by the reference data, where the travel times were rounded values (the times were multiples of five) rather than exact ones. The distribution of distances and times underlined the modal shares, with an almost uniform distribution between short trips (less than 2 km) performed on foot and for long trips (more than 2 km) mainly made by car. These last results validate our approach to reproducing mobility behaviors for our study case. Then, we evaluated the use of the park-and-ride (*PR*) facilities and the user profiles combining private cars and public transports.

#### 4.2.3. Assessment of the Use of *MEL PR*s and the Inter-Modal Users’ Profiles

*PR*s are the infrastructures required to combine private cars and public transport. Therefore, the choices of their location, capacity, and cost of use are key elements for this inter-modal practice. Based on the number of entries, we assess the capacity of *PR* facilities to absorb *PV*+*PT* transport demand. Table 2 provides information on the number of *PR* entries during the morning peak hours for the departures from home. The capacity of each *PR* (provided in parentheses) gives an idea of their filling rate.

During the peak hour, seven *PR*s (bold in Table 2) were overloaded. These *PR*s are mainly located in the two neighboring principal municipalities, particularly Lille (Les Près 1, Porte des Postes, CHU-Eurasanté, and Porte d’Arras) and Tourcoing (Gare de Tourcoing and Pont de Neuville). Apart from the particularity of being located in the main towns, these *PR*s have parking spaces (only 100 places on average). This small size is undoubtedly a strategy for developing *PR*s around primary cities to better manage flows for this area. However, the simulation shows that these *PR*s cannot contain the transport demand. Due to a lack of usage data, we cannot check this situation.

The profiles of individuals using *PV*+*PT* alternatives during their travels are also interesting. Figure 7 presents the sociodemographic distributions of agents combining *PV* and *PT* and their trip purposes.

The majority of the *PV*+*PT* users from the simulation and observed data were between 18 and 54 years old. This result is consistent with the age required to obtain a driver’s license and acquire a car. However, the simulation failed to reproduce the people aged between 18 and 24 years old well. This use rate of the *PV*+*PT* alternative is mainly explained by the low motorization (car ownership) for this category of agents in the synthetic population. Contrary to the reference situation, the simulation showed that men used the inter-modality more than women. In terms of professional status, the results of the simulation matched perfectly with the reference data. Workers practiced the inter-modality more. This result can be explained by the professional trips from the peripheral areas (characterized by car ownership) toward Lille (the main activity area) in particular. We also note a good result concerning the trip purposes: work, leisure, and other (e.g., visits). The educational motivation did not really give the desired result in the simulation, while that for shopping was overestimated. This was also justified by the motorized observations of the 18–24 year old group in the simulation.

The simulation successfully reproduced the observed behavior with *PV*+*PT* for the distances, durations, motivations, and socio-professional categories. The limitations of the simulation and the lack of data for *PR* uses will now be discussed.

## 5. Discussion

The anonymized data from different surveys (Section 5.1) and the interest in the obtained (more recent) data from the sensors (Section 5.2) are successively discussed.

### 5.1. Anonymized Data from Surveys

Our methodological approach aims to simulate realistic transportation scenarios from anonymized open data. The good results obtained by our algorithm in building the O/D coordinates depended on the area size and the number of points to be generated randomly. The larger the size of the drawing sector, the more random points should be generated to have a better chance of approximating the Euclidean distances. Therefore, the calculation times for searching for pairs of synthetic O/D points becomes very high, preventing other possibilities from being explored. One solution to this problem would be to use geographic databases (e.g., *BD TOPO*, *BPE*, or *OSM*) to generate a fixed set of O/D coordinates corresponding to potential activity locations (e.g., home, workplace, and school). The determination of the points thus generated would allow quickly inferring the pairs of coordinates by reducing the computational times.

Another point of discussion deals with the modeling of the choice for *PV*+*PT* alternatives. Its utility function is essentially based on the private vehicle and public transport. However, combining *PV* and *PT* for the same trip depends on park-and-ride facilities and requires the attributes of the infrastructure (such as the capacity, price, and accessibility). Thus, the utility function of *PV*+*PT* alternatives could be defined in terms of three components [42]:The *PV* characteristics between home and the *PR*;The specific utility of the *PR*, taking into account its own characteristics;The *PT* characteristics between the *PR* and the destination.

In addition, specific survey data on inter-modality could enrich the synthetic population with user behaviors and their trip purposes [4]. Unfortunately, to our knowledge, there are no such data for our country. Other data sources should be explored to have additional information on the inter-modality.

### 5.2. Data from the Sensors

As mentioned above, one of the main limitations for the inter-modal agent-based simulation is the quality of the available data. The data concerning these inter-modal practices are based on a few data sources. Reproducing more realistic simulations for *ABM*s remains a difficult problem. For example, usage data for *PR* facilities can be useful for calibrating the simulation model. The data can be obtained through the installation of access control devices for these different *PR*s (a method called passive data; see Section 2.1.2). These devices, equipped with sensors, can provide real-time feedback on *PR* use (e.g., number of entries and remaining places). In addition, the user’s sociodemographic data (e.g., age, gender, and working status) can come from card IDs. Such data (from a global perspective) can also be used to perform real-time simulations where agents update their decisions by considering the current traffic (e.g., congestion and availability of parking places in the *PR*s).

## 6. Conclusions

In this paper, we proposed studying the inter-modal travel behaviors from the perspective of the *ABM* and provide a generic framework to simulate the inter-modality from open data. In this context, we focused on the inter-modal combination of a private vehicle with public transport (*PV*+*PT*). The agent-based approach is suitable for studying these mobility behaviors, particularly the inter-modality. The modal choice for each agent was thus performed by a *multinomial logit* (*SI-MNL*) to select an alternative. This approach requires disaggregated data (census and household travel surveys) to generate agent-based transport scenarios and to reproduce travel behaviors as faithfully as possible.

This model was subsequently integrated into the well-known *MATSim*/*eqasim* simulation tool. We described the changes proposed for the definition of these new transportation modes and the inter-modal routing through the park-and-ride facilities (the main infrastructures allowing the combination of a private vehicle and public transport). Our model was also applied to an urban context (*MEL*). After the calibration and validation of the simulation to reproduce the inter-modality practices (modal shares, trip duration, and distance), we evaluated the use of the park-and-ride facilities (*PR*) by evaluating the number of entries and their capacities. We showed a consistency between the simulation outputs and the actual use of different *PR*s, particularly during peak hours. Due to the absence of data for the *PR*s, it was difficult to compare the simulation results with a reference situation. We also studied the user profiles and their motivations for inter-modal trips. The simulations reproduced the shares of individuals practicing inter-modality, such as for persons aged from 18 to 54 years.

As part of future work, we will plan to study the traffic congestion for the city center of Lille. The objective will be to investigate the travel behaviors under incentive (free access to *PR*s) and coercive (urban toll) constraints. The exploitation of data from certain typical modes may improve the results obtained by our approach, such as bus management [43] or bike-sharing services [44].

## Figures and Tables

**Figure 1 sensors-23-02540-f001:**
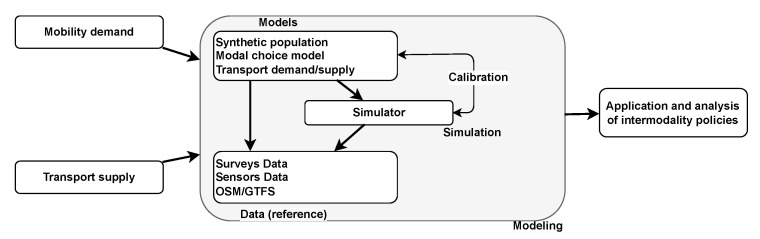
Diagram of the pipeline of an agent-based inter-modality simulation.

**Figure 2 sensors-23-02540-f002:**
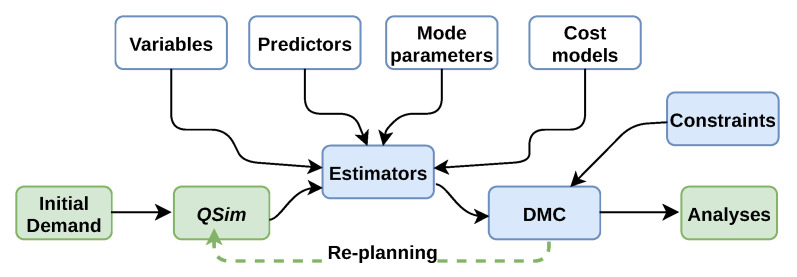
Simulation cycle for *MATSim* and *eqasim*- Adapted with permission from ref. [37].

**Figure 3 sensors-23-02540-f003:**
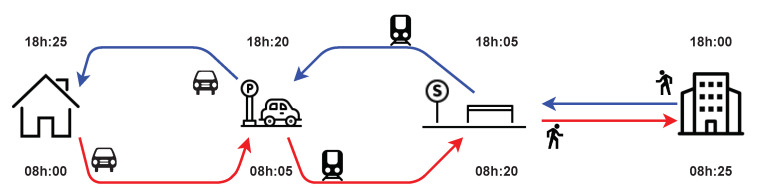
Inter-modal travel combining personal car and public transport, where *car_pt* mode for the outward journey is represented in red and *pt_car* mode for the return journey is in blue.

**Figure 4 sensors-23-02540-f004:**
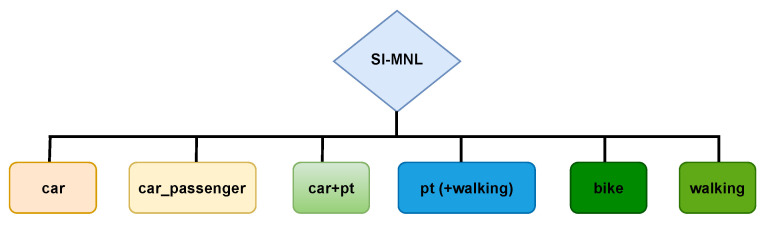
Structure of the *MNL* for modal choice estimation, considering the inter-modal alternative.

**Figure 5 sensors-23-02540-f005:**
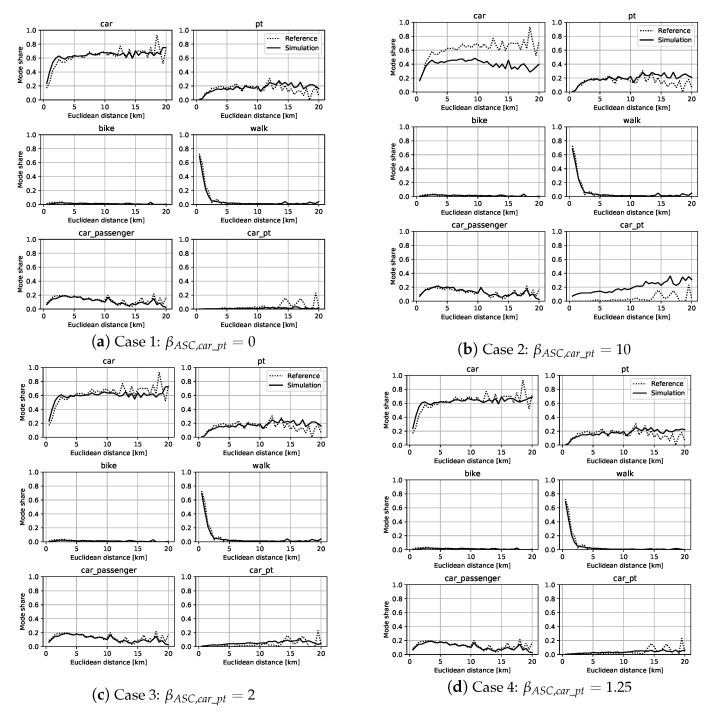
Modal shares during the calibration step for different values of $\beta_{ASC,car\_pt}$.

**Figure 6 sensors-23-02540-f006:**
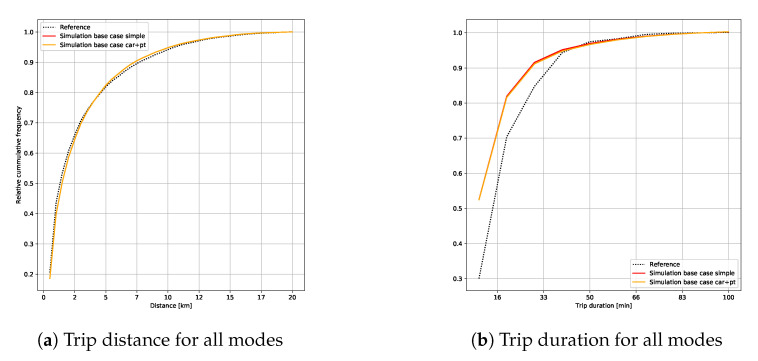
Distribution of trip distances and durations for all modes from simulation and reference.

**Figure 7 sensors-23-02540-f007:**
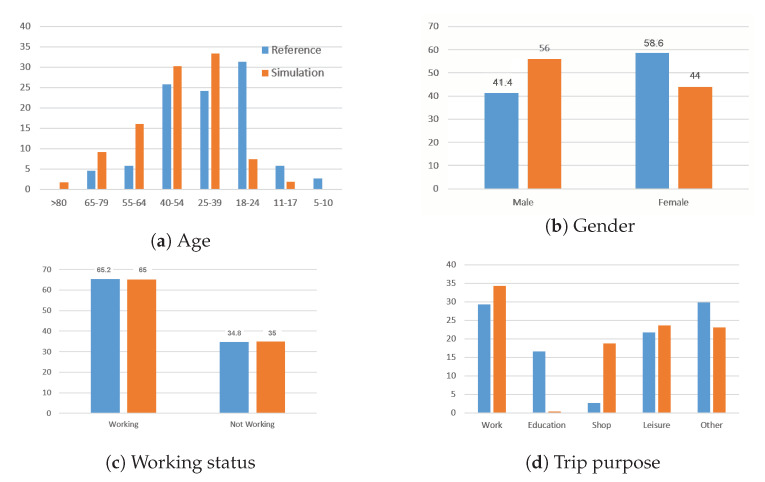
Sociodemographic characteristics of users of the *PV*+*PT* alternative and its trip purpose.

**Table 1 sensors-23-02540-t001:** Parameter values estimated with *Biogeme* software. Number of parameters: 13; sample size: 5.114; init log likelihood: −8668.555; final log likelihood: −5086.908; rho square: 0.412.

Alternatives	Parameters	Value	*p*-Value	Robust Std. Err.
*PV*	$\beta_{ASC,PV}$	−1.19	$3.77\times 10^{-15}$	0.132
	$\beta_{inVehicleTime,PV}$	−0.0051 min${}^{-1}$	0.591	0.00808
*PT*	$\beta_{ASC,PT}$	−2.24	0.0	0.103
	$\beta_{inVehicleTime,PT}$	0.00291 min${}^{-1}$	0.0618	0.00147
	$\beta_{accessEgressTime,PT}$	−0.0163 min${}^{-1}$	0.884	0.0414
	$\beta_{numberTransfers}$	−0.168	0.999	2.19
	$\beta_{transferTime,PT}$	−0.823 min${}^{-1}$	0.973	0.4
*Bicycle*	$\beta_{ASC,Bike}$	−1.64	0.0	0.205
	$\beta_{travelTime,Bike}$	−0.112 min^−1^	0.0	0.0147
	$\beta_{highAge,Bike}$	0.00925	0.0402	0.00484
*Walking*	$\beta_{ASC,walk}$	1.77	0.0	0.153
	$\beta_{travelTime,walk}$	−0.137 min^−1^	0.0	0.00618
*Other*	$\beta_{cost}$	−0.293 euro^−1^	$2\times 10^{-15}$	0.037
*Calibration*	$\theta_{parkingSearchPenalty}$	4	−	
	$\theta_{accessEgressWalkTime}$	4	−	−
	$\beta_{ASC,car\_pt}$	1.25	−	−

**Table 2 sensors-23-02540-t002:** Number of cumulative entries for the 12 *PR*s of *MEL* between 7:00 a.m. and 9:00 a.m.

*PR*	Number of Entries	*PR*	Number of Entries
Saint Philibert (330)	100	**Pont de Neuville (42)**	1390
**CHU-Eurasanté (250)**	690	**Armentières (450)**	730
**Les Près 1 (130)**	1370	Don Sainghin (467)	170
**Porte des Postes (90)**	2050	La Bassée (166)	90
**Porte d’Arras (100)**	780	Seclin (370)	130
**Gare de Tourcoing (90)**	1560	4 Cantons (2088)	910

## Data Availability

Publicly available datasets were analyzed in this study. They are introduced in the paper.

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
