# Peer review of "Agent-Based Approach for (Peri-)Urban Inter-Modality Policies: Application to Real Data from the Lille Metropolis"

_sensors, 2023, doi:10.3390/s23052540_

Round 1

Reviewer 1 Report

This manuscript tries to use Agent-based model to provide a simulation framework to address inter-modality policies at different scales. It provides a thought-provoking insight into the combination of spatial-temporal data and Surveys Data in urban inter-modality study. However, some flaws in the research should be modified.

1. The content of the second chapter of the article is too complex and lengthy, for example, the traditional four stages model is not quantitative comparative analysis in the following text, nor is the main used method in the article. There are also have similar problems in other sections of Chapter 2. It is suggested to be modified

2. In Figure 2 , Please explain why the PT+PT mode still exist? As mentioned above, the same PT mode will be merged.

3. In Section 4.2.1, the ASC value in the utility function is parameter allows capturing the variation of choice not explained by the known attributes exclusively. What is the basis for selecting four ASCs in Figure 5 of this section? In Figure 6, The two case curves for distance are completely overlapping.

4. The main technique contribution of this article is intermodal simulations based on MATSim, but how to use this simulation model in the case of experiences for the European Metropolis of Lille should be further explained.

5. The abstract of the article should introduce the data, methods and relevant conclusions of the article, which are rarely reflected in the current version, and it is recommended to modify this part to highlight the innovation of research methods or reveal new conclusions of the paper.

6. The conclusion part is relatively scattered and the key points are not prominent. It is suggested to highlight the main research conclusions and results obtained by the study, and list them one by one, without repeating the description of the work carried out.

Reviewer 2 Report

This study explores the inter-modal travel behavior of individuals from the perspective of an Agent Based Model. The paper is interesting to read, but there are key concerns that need to be addressed.

1.     Please employ the services of a proofreader to read and correct the grammatical errors in the document.

2.     Regarding the contributions, I think the first one is not a contribution. I do not see how building simulation configurations can be a contribution. I suggest that this be either removed or included elsewhere.

3.     The main concern of this Reviewer is why the Authors combined all the public transport modes as a single PT mode. Since these modes have unique features, it would be worthwhile to treat them as individual modes.

4.     Can the Authors include in Table 1 the P-values? This would help readers identify their level of statistical significance easily.

5.     What are the key implications of this research? Please discuss them in the paper.

Reviewer 3 Report

This paper proposed a simulation framework to address inter-modality policies at different scales to study the inter-modal travel behavior of individuals. The article is rich in content, but lacks organization. The authors should probably spend more time explaining the contribution of the paper. In addition, is there any theoretical innovation in this paper?

Round 2

Reviewer 1 Report

The authors have further modified and improved the paper based on the previous review opinions, which is generally closer to the standard of paper publication.

Generally speaking, the paper uses multi-source data to analyze travelers' travel choice behavior, the innovative contribution of main methods, and the model performance comparison with other existing methods, and proposes to make some further supplements in the conclusion section.

My opinion is that the author can basically meet the publication requirements of this journal after further checking and improving the description, English expression and charts of the innovative conclusions of the paper.

Reviewer 2 Report

Thank you for addressing my concerns.
